# Enhancing the energy level of regional digital innovation ecosystems: A configuration perspective

**Rong Huang, Shuai Mao** *

School of Economics and Management, Hubei University of Automotive Technology, Shiyan, Hubei, China

* 20120014@huat.edu.cn

## Abstract

This study introduces the concept of "energy level" into the analysis of innovation ecosystems. Drawing on the theory of "architects", we have identified the key architects of regional digital innovation ecosystems. By integrating Necessary Condition Analysis (NCA) and Fuzzy-set Qualitative Comparative Analysis (fsQCA), we examined the configuration effects of internal architects, external architects, and digital innovation habitats on the energy level of these ecosystems, utilizing data from 30 provinces in China as case studies. The results indicate that the contribution of a single architect to achieving a high energy level in a regional digital innovation ecosystem is limited and cannot be regarded as a necessary condition for driving a high energy level. However, core innovation actors and digital infrastructures emerge as more significant core conditions. The architects of regional digital innovation ecosystems exhibit multiple concurrent causal relationships. The configuration paths of high and non-high energy levels of ecosystems display a causally asymmetric "multiple paths to the same outcome" relationship. The three identified paths for driving high energy levels are categorized as "core actor-infrastructure"-driven paths. The findings of this paper hold substantial theoretical and practical significance for fostering the healthy development of regional digital innovation ecosystems.

**Data Availability Statement:** All the minimal anonymized dataset files are available from the Figshare database (DOI: 10.6084/m9.figshare. 25952083).

## 1. Introduction

The rapid development of digital technologies such as artificial intelligence, blockchain, cloud computing, and big data, acknowledged as novel innovation elements, is reshaping the landscape of innovation ecosystems. These technologies are merging with, restructuring, and optimizing traditional innovation elements, thereby transforming the original boundaries and contents of these ecosystems [1, 2]. This process is pioneering new avenues for value co-creation and fostering a multi-actor synergistic symbiosis at the ecosystem level l [3]. Concurrently, these advancements are enhancing the theoretical understanding and practical application of innovation ecosystems, culminating in the formation of the "digital innovation ecosystem" as an innovation paradigm [4, 5].

**Funding:** RH is supported by the Society Community Building Program of the Hubei Association of Higher Education (https://www.hahe.org.cn/) for the project "Research on the Collaborative Evolution Path of Innovation and Entrepreneurship Education Ecosystem in Local Universities" (Grant No.: 2023XD104). Additionally, RH is supported by the 2023 Annual Planning Project of the Commerce Statistical Society of China (http://www.china-cssc.org/) for the project "Research on the Configuration Path for the Stable Operation of Regional Innovation Ecosystems" (Grant No.: 2023STY43). SM is supported by the Hubei University of Automotive Technology's 2023 Undergraduate Course Teaching Case Library Construction Project (https://www.huat.edu.cn/) for the project "Teaching Case Library for Digital Marketing in the Automotive Industry" (Grant No.: XALK2023012).

**Competing interests:** The authors have declared that no competing interests exist.

A digital innovation ecosystem represents a multifaceted economic network composed of various entities and organizations leveraging digital technologies for product and service innovation [6, 7].

With the deep integration of regional economic development and digital innovation, the academic research perspective has gradually transitioned to the category of the regional digital innovation ecosystem. Numerous scholars have defined its connotation based on different perspectives, such as the symbiosis perspective and the complexity science perspective [8–10]. Combined with existing research, this paper posits that a regional digital innovation ecosystem refers to a complex, dynamic system characterized by the generation, application, and diffusion of digital innovation within a specific spatial and temporal context. This system operates within a digital ecological framework and functions through the competition and cooperation of digital innovation-related actors engaged in value co-creation. Compared with the traditional regional innovation ecosystem, the regional digital innovation ecosystem is supported by digital technology with convergence characteristics [11], which facilitates a closer symbiotic relationship of mutual synergy among constituent elements such as innovation actors and innovation resources. Furthermore, the driving effect of regional digital innovation ecosystems on regional innovation development has become more pronounced due to the enabling role and platform effect of digital innovation.

Within such ecosystems, particularly at a regional level, innovation actors are fundamental to innovation activities. They are pivotal in fostering the ecosystem's growth, acting as "architects" in its evolution [12, 13].

In exploring the architects of innovation ecosystems, existing research predominantly addresses the micro and meso levels. At the micro level, scholars concentrate on core enterprises, examining how technology, products, enterprise capabilities, and inter-firm interdependencies influence enterprise innovation ecosystems [14]. At the meso level, research often revolves around the "architects" theory, focusing on the formation and evolution of innovation ecosystems within the context of meso-industrial architecture [15]. Nonetheless, these studies typically view innovation ecosystems through an industrial lens, overlooking the spatial dimension and, fundamentally, constituting research on industrial innovation ecosystems. Cai and Yu [12] assert that innovation ecosystems inherently possess a spatial dimension, exemplified by regions such as Silicon Valley and Highway 128 high-tech zones in the United States, which are quintessential cases of regional innovation ecosystems. Compared to their industrial counterparts, regional innovation ecosystems are more prevalent. Concurrently, the advancement and implementation of digital technology have led to the emergence of structural features within digital innovation ecosystems, such as the digitization of innovation elements, the virtualization of participants, and the ecologicalization of participant relationships. These advancements persistently disrupt traditional economic models [16], thereby reshaping the modalities of value co-creation and sharing among innovation actors within the ecosystem. The impact of these actors on regional digital innovation ecosystems warrants further investigation. Consequently, drawing upon the "architects" theory, investigating the role of multiple interdependent innovation actors within regional digital innovation ecosystems is crucial. This exploration is key to uncovering the operational mechanisms of these ecosystems and boosting their energy levels.

The elevation of a regional digital innovation ecosystem's energy level involves the reconfiguration, integration, and linkage of its innovation elements to build innovation capacity, thereby augmenting the system's internal energy and external diffusion impact [17, 18]. Enhancing the ecosystem's energy level aids in optimizing regional resource allocation and spatial expansion capabilities, consequently improving the system's overall operational efficiency. In this context, several pertinent questions arise: What is the influence of innovation

actors, in their role as "architects," on elevating the energy level of regional digital innovation ecosystems? How do the effects differ among various innovation actors? To what extent do interactions between innovators impact regional innovation? How can regions effectively orchestrate innovation elements in line with the roles of innovation actors to boost the high-energy operation of digital ecosystems? Addressing these questions is currently of paramount importance.

A review of the literature reveals that the evolution of existing research on the "development paths of regional innovation ecosystems" has gone through three stages. The first posits a linear trajectory grounded in neoclassical economic theory and endogenous growth theory [19]. The second stage emphasizes the complexity of the evolutionary process of regional innovation systems, advocating systemic paths [20]. The third stage, inspired by the symbiotic evolution of natural biological populations, examines coupling paths through simulation analysis [21, 22].

Regarding methods to enhance the energy level of innovation ecosystems, scholarly perspectives converge on three main approaches. The first, from an enterprise perspective, focuses on enhancing collaboration, competition, and co-evolution among businesses and other organizations [16]. The second, from an industry viewpoint, considers the synergistic interactions between each innovation entity within the industry and its development environment [23]. The third approach, from a regional perspective, highlights the concept of clusters or communities, exploring the interaction between innovation groups and their environment [24]. While the first two perspectives concentrate on interactions among innovation actors, they lack insight into the influence of groups or clusters on the innovation ecosystem. Conversely, the regional perspective emphasizes the role of the group but does not fully explore the interactions among individual innovation actors. Unlike previous studies that used methods such as qualitative case analysis and quantitative regression, Qualitative Comparative Analysis (QCA) can study the causes of social phenomena by taking a "holistic" and "combinatorial" approach. It can identify the configuration paths leading to the outcomes of interest through differentiated permutations and combinations.

To address the research gaps mentioned above, this study will establish a framework for analyzing the architects of regional digital innovation ecosystems based on the "architects" theory. It will adopt a configurational perspective and utilize NCA along with fsQCA, which integrates quantitative and qualitative approaches, to analyze 30 provinces (cities, districts) in China. The aim is to explore the multiple simultaneous causal relationships and diverse configurational paths through which different combinations of influencing factors contribute to the elevation of the energy level of regional digital innovation ecosystems. The main findings are as follows:

First, individual architects are not prerequisites for achieving a high energy level in the regional digital innovation ecosystem. However, core innovation actors and digital infrastructures emerge as pivotal conditions. Numerous concurrent causal relationships interconnect core innovation actors, collaborative innovation actors, digital governance actors, regulatory innovation actors, digital infrastructure, and digital innovation environments, forming a complex web of interrelated components.

Second, the study reveals that the "core actor-infrastructure" pathway elevates the energy level of regional digital innovation ecosystems. A synergy between core innovation actors with high innovation levels and well-developed digital innovation infrastructure not only achieves a high energy level but also compensates for deficiencies in other antecedent conditions.

Third, digital governance actors are identified as pivotal bottlenecks influencing the energy level. The results of the NCA demonstrate that the presence of digital governance actors is the sole necessary condition to attain a 10% energy leve. This emphasizes the fundamental role of digital governance actors in boosting the energy level.

Finally, configuration pathways leading to a non-high energy level emphasize the importance of four key elements: core innovation actors, collaborative innovation actors, digital governance actors, and digital innovation environment. Without these, achieving a high energy level is unattainable, regardless of other conditions.

The structure of this article is as follows: Section 2 provides the theoretical foundation and research hypotheses, Section 3 outlines the research methodology, variable selection, and data sources, Section 4 presents and discusses the research findings, Section 5 summarizes the research conclusions and contributions, and provides further research directions.

## 2. Theoretical foundation and research hypothesis

### 2.1 Theoretical foundation

**2.1.1 "Architects" theory.** An "architect" is a pivotal entity that propels ecosystem development by establishing goals and orchestrating the actions of other participants [25]. As a system's most fundamental and distinctive element, architects, unlike general participating actors who only operate within their immediate business and technical domains, evolve collaboratively with common goals by coordinating the relationships between actors and members of the ecosystem. This leads to significant shifts in ecosystem structures. Theoretical discourse primarily focuses on the varying roles of architects in the formation and evolution of ecosystems [26, 27]. Table 1 compares the differences between the architects of the innovation ecosystem and general participants.

The foundational elements of an innovation ecosystem include actors, activities, locations, and relationships [27–29]. Actors within these ecosystems are typically categorized into core and non-core actors, based on their roles and influence [30, 31]. Scholarly consensus holds that core actors play a more pivotal role within the ecosystem. Many researchers have focused on understanding how different architects, particularly core actors, and dominant firms, influence the formation of industrial innovation ecosystems [28].

The study's findings indicate that core actors, as "architects," can occupy crucial bottleneck positions in the ecosystem. They achieve this through their foresight and institutional influence, by coordinating inter-actor relationships, controlling essential resources, facilitating information and resource flows, and consequently reshaping the ecosystem's architecture [32]. However, some scholars note that architects often engage in activities akin to public goods, such as building innovation networks and initiating innovation activities. This suggests that non-core actors might also assume the role of "architects" [12]. Furthermore, Jacobides, Cennamo and Gawer [33] contend that ecosystems do not arise spontaneously. They argue that architects could include not only core firms within the ecosystem but also entities originating from outside. The construction of innovation ecosystems entails the synergy of people, culture,

**Table 1. Comparison of differences between architects of the innovation ecosystem and general participants.**

|  | Role | Function | Scope of Influence | Control Capability |
|---|---|---|---|---|
| Architects | Core or non-core actors capable of influencing the construction and evolution of the entire innovation ecosystem. | Establishing ecosystem objectives and coordinating relationships to guide the evolution of the ecosystem | Global impact. Value creation and distribution patterns that influence the entire innovation ecosystem. | Having control over critical resources within the ecosystem, reshaping and leading the architecture of the innovation ecosystem. |
| General Participants | All members of the innovation ecosystem, including core companies, suppliers, partners, and others. | Within established frameworks and objectives, engaging in specific innovation activities and collaborations within their direct business and technical domains. | Local impact. Having an effect within their immediate business and technical domains, with limited influence on the overall architecture of the innovation ecosystem | Participating collectively in the operation and value creation of the innovation ecosystem, without the ability to alter the overall ecosystem architecture. |

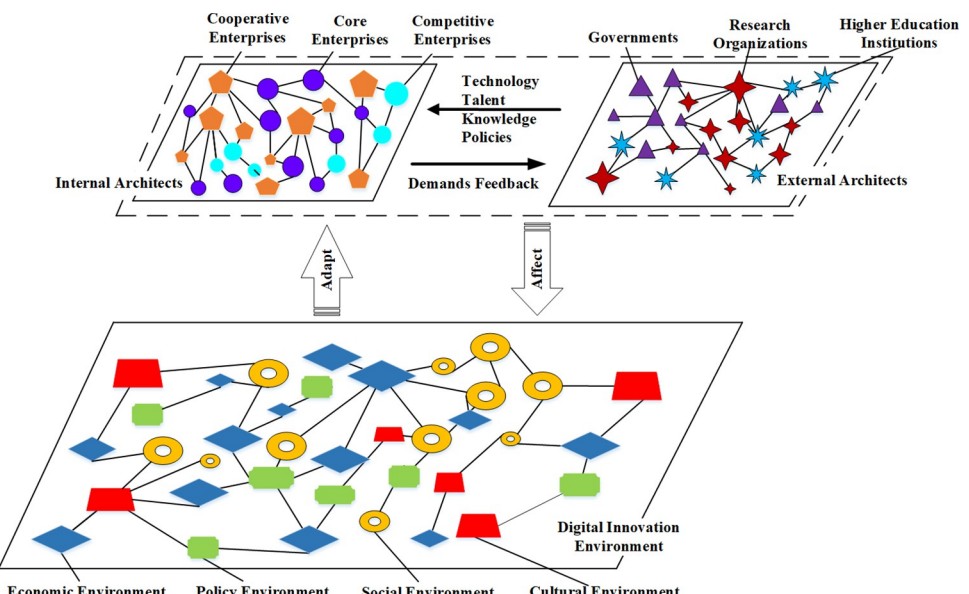

**Fig 1. Model of architects in the regional digital innovation ecosystem.**

and technology, along with the collaboration of academia, industry, and government. Building on this, some scholars have emphasized that the innovation ecosystem is an organic entity, emerging from the interaction of various participating actors and their social and natural environments. This interaction aims to elevate the level of regional innovation activities [34, 35].

Existing research on innovation ecosystems utilizing the "architects" theory predominantly focuses on the industry level, often overlooking the spatial dimension of these ecosystems [26]. It is important to recognize that while innovation ecosystems represent a specific form of industrial architecture, regional innovation ecosystems constitute a major category, as illustrated by various high-tech industrial clusters [12]. Although regional innovation ecosystems are more prevalent than their industrial counterparts, the application of the "architects" theory at the regional level has been relatively limited in scholarly research.

Consequently, building on existing literature, this paper extends the "architects" theory within the context of the innovation ecosystem. It further delineates the architects of regional digital innovation ecosystems into internal and external categories. As previously indicated, these architects do not operate in isolation; rather, they must engage in resource exchange and integration within the regional digital innovation habitat to achieve dynamic equilibrium and sustainable development of the ecosystem. In line with this, the paper presents a model of the regional digital innovation ecosystem composition, as illustrated in Fig 1.

**2.1.2 Energy level of regional digital innovation ecosystems.** The concept of "Energy Level" has its origins in physics, where it initially referred to the energy an electron possesses and its corresponding level. Over time, this term has been adapted to describe the degree of aggregation and radiative influence of a specific function. In this context, "Energy" signifies the magnitude and direction of energy, essentially the "amount of work" involved. "Level" represents the connections, hierarchy, and status within a substance or system, highlighting the relationships and relevance within the substance or system.

Currently, the "Energy Level" theory has broadened its application to various fields, including ecology and economics. Its definition concentrates on two main aspects. First, it adheres to the physical definition of energy level, emphasizing the magnitude of energy within entities

and their corresponding levels and statuses. Second, the theory expands its scope by integrating the economic characteristics of regions and industries, highlighting both the inherent energy of entities and their external radiative effects. In the academic realm, there is no universally accepted definition of the energy level of regional innovation ecosystems. Yang, Liu and Wei [18] proposed that the energy level of a regional innovation ecosystem primarily reflects the operational level of the system. They defined "Energy" as the increase in the overall system's innovation efficiency, and "Level" as the sustainable operation of the system. Building on this, Li and Rao [36] and Li, Rao and Yuan [17] posited that the energy level of regional innovation ecosystems represents a unification of the system's internal ecological structure and its external radiation function. This encompasses both "Energy Level Structure" and "Energy Level Effect." The "Energy Level Structure" denotes the energy inherent in the ecosystem's internal structure, whereas the "Energy Level Effect" refers to the flow of innovation elements among diverse entities within the ecosystem, driven by the differential energy potentials of various innovation actors. This dynamic results in both internal and external linkages, fostering diffusion and radiative interactions among the participants.

This paper builds upon the research of Li and Rao [36] and Li, Rao and Yuan [17], categorizing the energy level of regional digital innovation ecosystems into two dimensions: "energy level structure" and "energy level effect." Drawing from a comprehensive analysis of existing literature, we elucidate that "energy level structure" represents the energy of the regional digital innovation ecosystem at a specific time or period. This encompasses the ecosystem's influence or control over regional development, its contribution degree, and industrial competitiveness. The "energy level effect" pertains to the diffusion effect arising from linkages within or between various systems of the regional digital innovation ecosystem [37].

## 2.2 Research hypothesis

**2.2.1 Internal architects.**   Existing research indicates that interactions among system components are crucial for the formation and evolution of the innovation ecosystem [26]. Internal architects, primarily comprising digital core enterprises and supported by competing and participating enterprises, along with digital extension industries, constitute the main entities directly involved in creating value for digital products. These entities supply the requisite energy for the stable development of the digital innovation ecosystems.

Digital innovation ecosystems possess a richer array of innovation elements than traditional ecosystems, owing to advancements and enhancements brought by data and digital technologies [34]. Within specific industrial chains, digital core enterprises create comprehensive product-to-value chains. They do this by selecting and integrating enterprises or products that offer advanced technology or cost advantages [38]. This approach maximizes their digital innovation capabilities, generates radiative impacts, and facilitates their evolution into core innovation actors. Eventually, these enterprises develop into internal architects of the digital innovation ecosystem. Some scholars argue that core companies are the most critical innovation entities within the innovation ecosystem, capable of accelerating the innovation process, driving value realization, and promoting the development of the innovation ecosystem [39, 40].

The rise of competition instigates an unseen internal drive among collaborative innovation actors such as competing, participating, and core firms within the ecosystem, attracting high-quality innovation resources and amplifying inter-firm competition. To swiftly adapt to digitalization trends, enterprises embedded in the energy cluster must consistently strengthen their "endogenous force." This entails a continual enhancement of their competitiveness through product iteration, innovation, or upgrading, thereby contributing significantly to the

advancement of the regional digital innovation ecosystem [41]. For instance, enterprises specializing in digital technology address digital transformation challenges by advancing digital technologies. These enterprises pervade the entirety of industrial production and operation, offering essential services such as digital facilities, information, and support for all facets of production's digital transformation. Within the innovation ecosystem, they assume a pivotal role in collaborative innovation. Santoro, Bresciani and Papa [42] argue that the greater the distance between participants' knowledge repositories, the more it fosters innovation. Zang, Wang and Zhou [31] suggest that non-core enterprises play an indispensable role as participants in the ecosystem, contributing to functional complementarity with core enterprises and the enhancement of the ecosystem. We hypothesize:

*Hypothesis 1*: Core innovation actors can effectively enhance the energy level of regional digital innovation ecosystems.

*Hypothesis 2*: Collaborative innovation actors play a significant role in enhancing the energy level of regional digital innovation ecosystems.

**2.2.2 External architects.**   External architects predominantly include government bodies, universities, and research institutions. Though not directly participating in product value creation, they offer vital, ongoing dynamic support for the digital innovation ecosystem's development and serve a moderating function among innovation actors [26, 41].

The government plays a dual role in the regional digital innovation ecosystem. Firstly, it acts as a strategy maker, guiding regional digital development and moderating the collaboration among various innovation actors within the internal architect energy cluster. Secondly, the government is a supporter and regulator of the ecosystem's functioning, providing necessary digital infrastructure and policy support to foster regional digital advancement [12]. Moreover, in digital innovation ecosystems, effective governance using digital technology significantly enhances the efficiency and level of services provided by regional government platforms [43]. Overall, the government's role is pivotal as a governor in the establishment and maintenance of regional digital innovation ecosystems. Some scholars argue that governments inject public value creation capabilities into the digital economy by improving digital transformation governance systems, motivating physical enterprises to understand and implement digital innovation, and accelerating digital transformation [44, 45].

Universities and research institutes, leveraging their resources like elite research teams, leading innovators, advanced experimental equipment, and cutting-edge information [38], play a crucial role in the regional digital innovation ecosystem. They engage in basic research and development of forefront technologies pertinent to digital advancement, continuously producing digital talents, technologies, and other vital "energies." These outputs circulate within the region, fostering innovative outcomes [16], and contributing to overcoming significant bottlenecks in regional digital technology development. By applying digital means and theoretical knowledge practically, they maximize the overall benefits for the region. Scholars such as Owen, Vedanthachari and Hussain [46], Valavanidis [47], Castañón and Bustamante [48] argue that in the context of rapid digital application innovation and research and development challenges, universities and research institutions can exert a pivotal regulating and supporting influence through knowledge spillover. We hypothesize:

*Hypothesis 3*: Digital governance actors can effectively enhance the governance level of regional digital innovation ecosystems.

*Hypothesis 4*: Regulating innovation actors play an important moderating and supporting role in the enhancement of the energy level of regional digital innovation ecosystems.

**2.2.3 Digital innovation habitat.** The concept of a digital innovation habitat encompasses the aggregate of external conditions that impact the emergence, survival, and evolution of digital innovation ecosystems [10]. These conditions exert a direct or indirect influence on the effectiveness and sustainability of innovation ecosystems. A conducive innovation habitat is instrumental in expediting the development of digital innovation ecosystems.

Digital infrastructures, serving as both software and hardware support for digital technology, are a critical element for the realization of modular characteristics and a key environmental factor in the existence and development of the regional digital innovation ecosystem. It primarily encompasses infrastructures characterized by public good attributes, including data centers, smart computing centers, technology centers, and 5G base stations [38]. Digital infrastructure facilitates the transmission, interaction, sharing, development, and utilization of data and information resources among various innovation actors. This, in turn, aids both internal and external architects of the innovation ecosystem in realizing value creation and interactive coordination. Bejjani, Göcke and Menter [49] argue that digital infrastructure enables complementary development among entities in the ecosystem. Makori [50] suggests that digital infrastructure and information technology construction can promote digital transformation and sustainable development of the innovation ecosystem.

Digital technology and ecosystems enhance the intricate interconnections among innovation actors and between systems, thereby enriching the innovation environment within the ecosystem. Influenced by the digital innovation environment, each actor cooperates and interacts through distinct symbiotic patterns, establishing a symbiotic network to collectively engage in digital innovation activities [10, 51]. Vaidian, Jurczuk [52] suggest that the digital innovation environment supports other stakeholders through its conceptual framework, technological environment, and physical infrastructure. Liu, Cheng and Ayangbah [53] believe that fostering a better digital innovation environment helps drive businesses to intensify their efforts in digital innovation, resulting in significant benefits. We hypothesize:

*Hypothesis 5*: Digital infrastructures provide platform support for the elevation of the energy level of regional digital innovation ecosystems.

*Hypothesis 6*: The digital innovation environment provides environmental security for the enhancement of the energy level of regional digital innovation ecosystems.

## 3. Materials and methods

### 3.1 NCA and fsQCA

In contrast to traditional regression analysis, which emphasizes the "net effect" of individual factors, Qualitative Comparative Analysis (QCA) employs multiple case studies to explore the equivalency of different antecedent states through causal inference using Boolean algebra logic. This methodology includes Clear-set Qualitative Comparative Analysis(csQCA), Fuzzy-set Qualitative Comparative Analysis(fsQCA), and Multi-valued Qualitative Comparative Analysis(mvQCA).

The fsQCA utilizes a fuzzy membership calibration method to transform data into any numerical representation within the [0–1] interval, effectively avoiding the loss of data information during processing. It can be applied not only to categorical sample data but also to continuous data [54]. This method is capable of handling problems involving partial membership relationships more effectively. The fsQCA method can detect multiple concurrent causal relationships and reveal the complex relationships among various actors' influences on the elevation of the energy level of regional digital innovation ecosystems from a holistic perspective

[54]. It identifies "which configurations of conditions lead to the occurrence of the outcome variable, and which configurations of conditions lead to the non-occurrence of the outcome variable." which perfectly aligns with the multi-path research process of elevating the energy level of regional digital innovation ecosystems. Therefore, fsQCA is utilized for the empirical analysis in this study.

Although fsQCA is adept at qualitatively determining the necessity of antecedent conditions for an outcome, Necessary Condition Analysis (NCA) offers a more expansive analysis. NCA tests the necessity of conditional variables by employing two techniques: the Ceiling Envelope with Free Disposal Hull (CE-FDH) and Ceiling Regression with Free Disposal Hull (CR-FDH) [55]. This approach not only identifies the necessary conditions leading to the outcome variable from a quantitative perspective but also quantifies the magnitude and bottleneck degree of these necessary conditions. Hence, integrating NCA with fsQCA can significantly bolster the persuasiveness and scientific validity of research findings [55].

## 3.2 Data collection

This study focuses on the 30 provinces (cities, districts) in China (excluding Hong Kong, Macao, Taiwan, and Tibet) as its research subjects, considering data availability. Incorporating the regional attributes of the regional digital innovation ecosystem, each province (city, district) is treated as a distinct regional digital innovation ecosystem, serving as a research case. To mitigate the singularity of cross-sectional data and the impact of random disturbances, the study utilizes the average values of each indicator over a three-year period. For antecedent conditions, the paper selects the average values of the indicator measurement data for the years 2018, 2019, and 2020. Considering the time lag effect, the outcome variable is selected by aligning the average values of the indicator measurement data for the years 2020, 2021, and 2022. Missing data were imputed using the mean value. The relevant indicators for the energy level structure and energy level effect are sourced from the "China Digital Ecology Index Report," while data for other indicators are obtained from reliable sources such as the "China Science and Technology Statistical Yearbook," "China Statistical Yearbook," "China High-Tech Industry Statistical Yearbook," "China Regional Innovation Capacity Assessment Report," "White Paper on the Development and Application of Industrial Internet," the China Internet Network Information Center, and the National Bureau of Statistics.

## 3.3 Variable descriptions

**3.3.1 Outcome variable.** This paper builds upon existing research on regional innovation ecosystems, integrating the concept of digital innovation ecosystems. It focuses on the goal of enhancing the energy level of the ecosystem, which is categorized into two dimensions: energy level structure and energy level effect [17, 18, 36].

*Energy Level Structure.* As mentioned earlier, "Energy Level Structure" refers to the influence, control, contribution, and competitiveness of the digital innovation ecosystem on regional economic development. Referring to existing literature, this paper uses the "Digital Capability Index" from the China Digital Ecology Index Report to measure [56]. According to the report, digital capability reflects the transformation stage of digital development and is a key driver for the evolution of regional digital ecosystems. The level of digital capability directly determines whether a region can achieve leapfrog development. It is also the backbone of the regional digital ecosystem and a necessary condition for its solid development, enabling data to realize its information value. This includes three sub-indicators: Digital Talent, Technological Innovation, and Digital Security, which collectively ensure a degree of scientific comprehensiveness [57].

*Energy Level Effect.* Due to the ecological energy potential difference between different systems, the overflow of energy generated by innovative elements such as data, labor, capital, and technology in the regional digital innovation ecosystem becomes a supplement to the material and energy in the innovation chain. This energy diffusion promotes a closely linked network between different innovation ecosystems. Stronger network connectivity enhances the ability of digital innovation actors to achieve breakthroughs through knowledge search and integration [58]. This study selects the "Data Resource Index" from the China Digital Ecology Index Report, reflecting data flow between systems, to measure the energy level effect of the regional digital innovation ecosystem [56]. This index comprises data openness and data circulation, capturing the development level of data elements within the digital ecosystem. It specifically reflects progress in areas of openness, sharing, circulation, and trade [57].

**3.3.2 Antecedent conditions.** In regional digital innovation ecosystems, the intangible nature of digital innovation actors and the dynamic nature of their relationships make it challenging to derive precise conclusions based solely on quantitative or diversity metrics [59, 60]. Consequently, this paper adopts the methodology of Li and Rao [10] to assess the specific performance of internal and external architects at the industry level. This is achieved by analyzing the operation of the digital industry sector and the digital application within the non-digital industry sector in the national economy.

*Core innovation actors.* The regional digital innovation ecosystem has dual attributes of both regional and industrial characteristics, yet comprehensive data statistics on the digital industry are currently lacking. Based on the findings of Li and Rao [10], this study utilizes data on software industry revenue, information technology industry revenue, and embedded system software revenue from the China Statistical Yearbook as metrics for evaluating core innovation actors.

*Collaborative innovation actors.* Competing and participating companies within the regional digital innovation ecosystem, along with core innovative actors, collaborate to bring high-quality innovation resources to the ecosystem [41]. This collaboration promotes regional or industry-specific digital transformation and development, while also indirectly stimulating other traditional industries Li and Rao [10]. Drawing from existing research on the digital economy, regional innovation ecosystems, and regional digital innovation ecosystems [10, 61, 62], this study adopts the Digital Spillover Index, Digital Convergence Index, and Digital Inclusive Finance Index from the China Digital Economy Index Report and the Digital Inclusive Finance Index as metrics to evaluate collaborative innovation actors [10]. These indices are chosen to represent the collaborative innovation contributions of non-core innovation actors like competing and participating enterprises within the ecosystem's architectural framework.

*Digital governance actors.* Previous studies have mostly measured the government's digital governance capabilities using indicators such as the number of government websites, the number of government agency Weibo accounts, the number of government headline accounts, and the number of governments TikTok accounts. To reflect the level of government digital governance more comprehensively, this paper utilizes the Digital Government Index from the China Digital Ecology Index Report as a metric. This index evaluates the digital governance of governments across four dimensions: organization, institutional system, governance capacity, and governance effect [56].

*Regulating innovation actors.* Universities and research institutions are the regulating innovation actors in the regional digital innovation ecosystem, driving regional development through knowledge spillover. Digital startup companies can enhance their digital innovation capabilities and outcome conversion rates via technological cooperation with these institutions. Additionally, startups can strengthen their human resources by attracting talent.

Informed by existing literature, this study selects the number of institutions of higher education, the number of high-tech enterprises, and the number of scientific research institutions., as listed in the China Statistical Yearbook, to measure the presence of regulating innovation actors [16, 38].

*Digital innovation infrastructures.* Based on the indicator analysis in the China Digital Ecology Index Report, this paper selects the "Innovation Infrastructure Index" as a measurement indicator. This index comprises two aspects: the New Infrastructure Competitiveness Index and the Cloud Computing Index. The New Infrastructure Competitiveness Index gauges the development level of new infrastructures in each region, focusing on network infrastructure, new application infrastructure, and new industry infrastructure. Meanwhile, the Cloud Computing Index primarily reflects the extent of cloud adoption and the development process in each region [56, 57].

*Digital innovation environment.* The innovation environment encompasses policy, economic, social, and cultural environments [63], within which all innovation activities by both internal and external architects of the regional digital innovation ecosystem occur. Drawing from existing research, the policy environment is assessed using two indicators: the proportion of education expenditure and the proportion of fiscal expenditure on science and technology, indicating the local government's focus on talent cultivation and R&D [38, 64]. The economic environment is measured by per capita regional gross domestic product. The social environment is gauged by the comprehensive index of the business environment [65]. Lastly, the cultural environment is evaluated using the indicator of average years of education per capita [9, 24].

From a configurational perspective, the impact of internal architects, external architects, and digital innovation habitats on the energy level of regional digital innovation ecosystems is interdependent, exhibiting coordination and synergy. The effects of various antecedents can either be amplified through mutual adaptation or counterbalanced through substitution. Consequently, this paper presents an analytical framework, as depicted in Fig 2.

## 3.4 Calibration

For the outcome variables and each antecedent condition, this study employs the entropy method to ascertain the weights of each indicator and compute the composite score for each case. Concurrently, the direct calibration method is applied to calibrate both the outcome

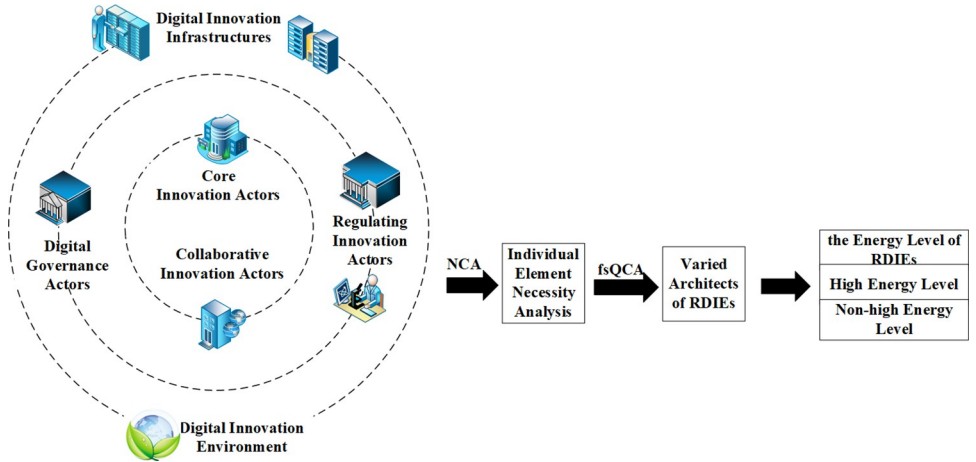

**Fig 2. The analytical framework.**

**Table 2. Calibration anchors of each fuzzy set.**

| Antecedent Conditions | Full Membership | Crossover Point | Full Non-membership |
|---|---|---|---|
| Energy Level | 0.024337 | 0.014014 | 0.005411 |
| Core Innovation Actors | 0.046206 | 0.009939 | 0.001155 |
| Collaborative Innovation Actors | 0.021837 | 0.016545 | 0.00988 |
| Digital Governance Actors | 0.051384 | 0.038843 | 0.022182 |
| Regulating Innovation Actors | 0.022405 | 0.013847 | 0.006927 |
| Digital Innovation Infrastructures | 0.063137 | 0.033495 | 0.023697 |
| Digital Innovation Environment | 0.05935 | 0.037866 | 0.029 |

variables and antecedent conditions. The three anchor points—full membership, crossover point, and full non-membership—are primarily determined using the upper quartile, median, and lower quartile of the probability density function of each variable within the sample range [18, 66]. The specific calibration anchors for the outcome variables and antecedent conditions are detailed in Table 2.

## 4. Analysis and results

### 4.1 Necessary condition analysis

NCA is adept at delineating the types and degrees of necessary conditions, aiding in addressing the question, "What is the minimum level of antecedents required for a regional digital innovation ecosystem to achieve a specific energy level?" However, the approach to analyzing necessary conditions in NCA fundamentally differs from that in fsQCA, often leading to NCA identifying a greater number of necessary conditions than fsQCA. Consequently, these methods provide complementary insights. Thus, it is important to recognize that NCA should not be employed merely as a tool for robustness testing in QCA's necessary condition analysis [55].

NCA utilizes a unique approach called the Ceiling Line in an $X-Y$ plot to distinguish between observable and unobservable regions. This technique helps in determining the necessity of a condition variable by checking for the existence of a blank region above the Ceiling Line [55, 67]. NCA typically employs two main techniques for cap analysis: Ceiling Envelope with Free Disposal Hull (CE-FDH) and Ceiling Regression with Free Disposal Hull (CR-FDH) [67]. The choice between these two techniques depends on the nature of the conditions and outcomes being analyzed. Specifically, when the conditions and outcomes are dichotomous variables or discrete variables with less than five level classes, the CE-FDH method is used for calculation. In cases where the variables do not fit this criterion, the CR-FDH method is employed.

In this paper, two analytical methods, CE-FDH and CR-FDH, are employed to construct an $X-Y$ scatterplot with combing parameters. Fig 3 in the paper displays this scatterplot, including the ceiling line. The size of the space in the upper left corner of the plot, relative to the total space of observations, indicates the degree to which variable $X$ (the condition) constrains variable $Y$ (the outcome) [55]. Essentially, the larger this space is, the greater the constraining influence of $X$ on $Y$.

Additionally, the outcomes of NCA are presented in Table 3.

To determine necessary conditions using NCA, two criteria must be met: Firstly, an effect size $d \geq 0.1$ is required, where a value between 0.1 and 0.3 indicates a low impact and a value between 0.3 and 0.5 suggests a high impact. Secondly, the results of Monte Carlo simulations

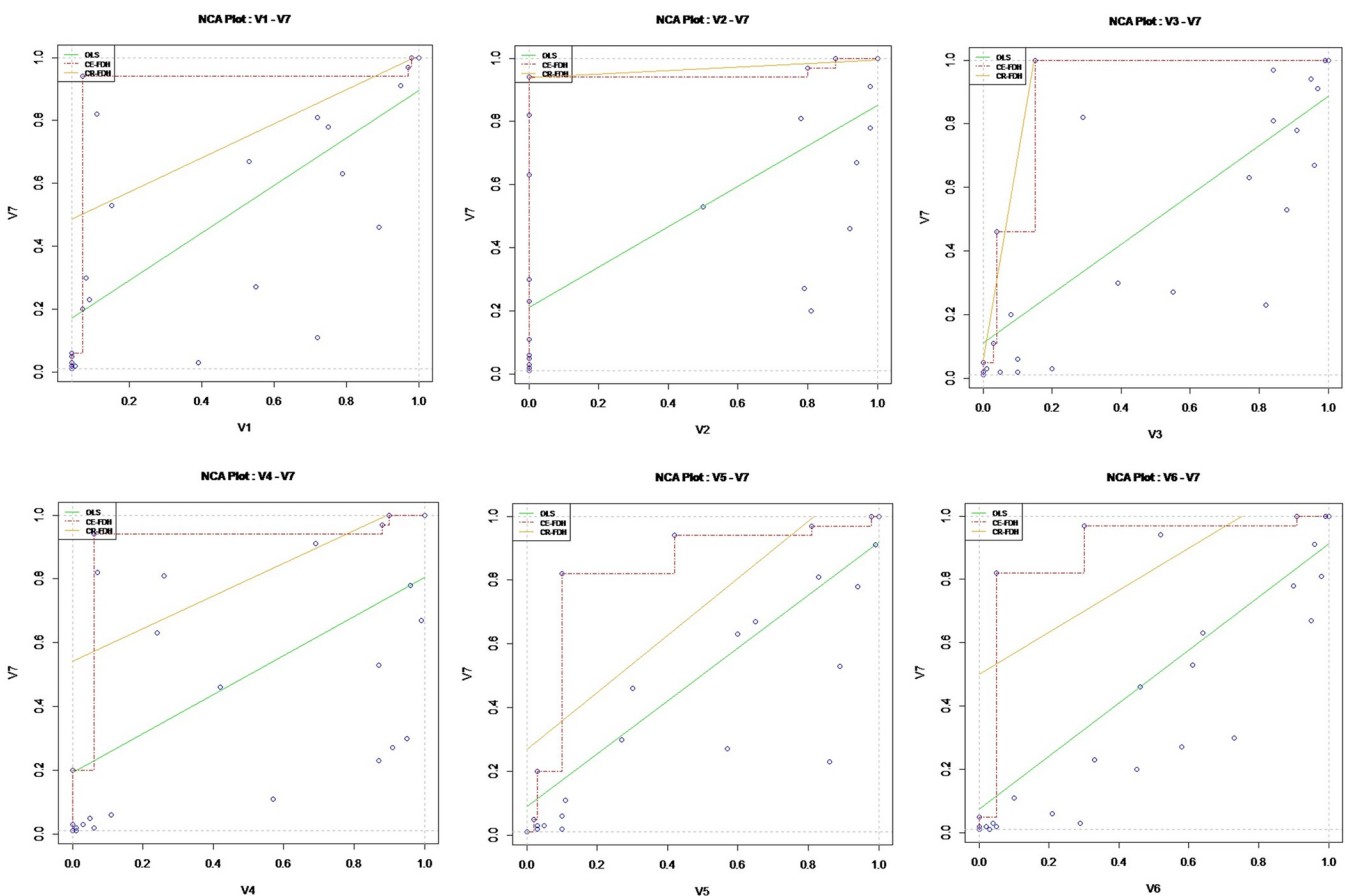

**Fig 3. Scatter plots with ceiling lines.** CR-FDH generates straight lines (i.e., solid lines), and CE-FDH generates the piecewise lines (i.e., dashed lines).

of permutation tests need to be significant ($p \leq 0.01$), with a p-value of 0.01 or less [55]. Based on these criteria, this paper identifies the necessary conditions for enhancing the energy level of RDIEs. As detailed in Table 2, core innovation agents, regulating innovation actors, digital innovation infrastructures, and digital innovation environment are identified as necessary

**Table 3. Results of NCA.**

| Antecedent Conditions | Methodology | C-accuracy | Ceiling Zone | Scope | Effect Size | P-value |
|---|---|---|---|---|---|---|
| Core Innovation Actors | CR | 90% | 0.245 | 0.95 | 0.257 | 0.000 |
| | CE | 100% | 0.083 | 0.95 | 0.087 | 0.000 |
| Collaborative Innovation Actors | CR | 76.7% | 0.033 | 0.99 | 0.034 | 0.001 |
| | CE | 100% | 0.050 | 0.99 | 0.021 | 0.001 |
| Digital Governance Actors | CR | 96.7% | 0.070 | 0.99 | 0.071 | 0.003 |
| | CE | 100% | 0.097 | 0.99 | 0.098 | 0.000 |
| Regulating Innovation Actors | CR | 86.7% | 0.205 | 0.99 | 0.207 | 0.000 |
| | CE | 100% | 0.098 | 0.99 | 0.099 | 0.000 |
| Digital Innovation Infrastructures | CR | 93.3% | 0.299 | 0.99 | 0.302 | 0.000 |
| | CE | 100% | 0.171 | 0.99 | 0.173 | 0.000 |
| Digital Innovation Environment | CR | 90% | 0.188 | 0.99 | 0.190 | 0.000 |
| | CE | 100% | 0.111 | 0.99 | 0.112 | 0.002 |

conditions for achieving a certain level of energy level enhancement in regional digital innovation ecosystems. Specifically, digital innovation infrastructure emerges as a necessary condition with a high level of impact ($d>0.3$), while core innovation agents, regulating innovation actors, and digital innovation environment are necessary conditions with a medium level of impact ($d>0.1$).

The necessity of an antecedent condition for the outcome variable is not static, hence the need for further analysis of the bottleneck level. The bottleneck level indicates the minimum percentage level of the antecedent condition $X$ that must be met for the outcome variable $Y$ to reach a specific level. In this context, the necessary condition acts as a bottleneck for the existence of the outcome, meaning that its absence cannot be compensated for by any other condition [55]. Given that all variables in this study are continuous, the bottleneck level is measured using the CR-FDH estimation method. In this framework, "NN" (Not Necessary) denotes that the antecedent condition $X$ is not necessary for achieving a particular level of the outcome variable $Y$, implying that any value of $X$ could potentially result in the desired level of $Y$, "NA" (Not Applicable) signifies that the value of $X$ cannot be calculated for the analysis. According to Dul, Hauff and Bouncken [55], a possible explanation is that the maximum possible value of the condition for the particular level of $Y$ according to the ceiling line is lower than the observed maximum value. In this paper, we address instances where "NA" appears in NCA by substituting it with the highest observed value (100.0%) of the variable X2. This adjustment can be done with the argument cutoff = 1 in the nca_analysis function, as recommended by [68].

Table 4 presents a detailed analysis of six antecedent conditions essential for examining the necessary relationships in varying degrees. Additionally, it quantifies the bottleneck effect size of these necessary conditions by specifying the minimum proportion of the conditional variable $X$ that must be met for the outcome variable $Y$ to achieve a particular percentage level within the observed range. For instance, to attain a 10% energy level in the total observational range, only the digital governance actor needs to achieve a minimum of 0.9%, rendering other conditions unnecessary. This underscores the foundational role of digital governance in driving the energy level within a regional digital innovation ecosystem. To reach a specific energy level, the criteria outlined in the bottleneck table must be satisfied; failure to meet these requirements results in the inability to achieve the desired energy level in the ecosystem. For example, a digital innovation environment above 30.4 is essential for surpassing a 70% energy level. When the digital innovation environment is at 30.4%, the highest achievable energy level of the regional digital innovation ecosystem is 70%.

**Table 4. Bottleneck level analysis of the NCA method.**

| Y | 1 | 2 | 3 | 4 | 5 | 6 |
|---|---|---|---|---|---|---|
| 0 | NN | NN | NN | NN | NN | NN |
| 10 | NN | NN | 0.9 | NN | NN | NN |
| 20 | NN | NN | 2.4 | NN | NN | NN |
| 30 | NN | NN | 4.0 | NN | 4.4 | NN |
| 40 | NN | NN | 5.5 | NN | 15.4 | NN |
| 50 | 3.9 | NN | 7.1 | NN | 26.5 | 0.6 |
| 60 | 22.9 | NN | 8.6 | 12.1 | 37.5 | 15.5 |
| 70 | 41.9 | NN | 10.2 | 31.4 | 48.5 | 30.4 |
| 80 | 60.9 | NN | 11.7 | 50.8 | 59.6 | 45.4 |
| 90 | 79.9 | NN | 13.2 | 70.1 | 70.6 | 60.3 |
| 100 | 98.9 | 100% | 14.8 | 89.5 | 81.7 | 75.2 |

**Table 5. Analysis of necessary conditions of fsQCA method.**

| Antecedent Conditions | High Energy level | | Non-High Energy level | |
|---|---|---|---|---|
| | Consistency | Coverage | Consistency | Coverage |
| Core Innovation Actors | 0.814516 | 0.857143 | 0.285714 | 0.305516 |
| ~Core Innovation Actors | 0.340054 | 0.319042 | 0.866402 | 0.825977 |
| Collaborative Innovation Actors | 0.756048 | 0.840807 | 0.224868 | 0.254111 |
| ~Collaborative Innovation Actors | 0.329301 | 0.294826 | 0.859127 | 0.781588 |
| Digital Governance Actors | 0.847446 | 0.846309 | 0.281746 | 0.285906 |
| ~Digital Governance Actors | 0.284946 | 0.280795 | 0.848545 | 0.849669 |
| Regulating Innovation Actors | 0.779570 | 0.777480 | 0.341931 | 0.346515 |
| ~Regulating Innovation Actors | 0.344758 | 0.340186 | 0.780423 | 0.782493 |
| Digital Innovation Infrastructures | 0.873656 | 0.883753 | 0.335317 | 0.335762 |
| ~Digital Innovation Infrastructures | 0.298387 | 0.290386 | 0.849868 | 0.862416 |
| Digital Innovation Environment | 0.862231 | 0.849669 | 0.282407 | 0.290279 |
| ~Digital Innovation Environment | 0.325941 | 0.325503 | 0.886905 | 0.877044 |

The fsQCA 3.0 software was utilized to analyze the necessary conditions for achieving high and non-high energy levels. Typically, the consistency threshold was established at 0.90, with antecedent conditions exceeding this threshold considered necessary. However, as indicated in Table 5, the consistency of all antecedent conditions fell below 0.9, suggesting none were necessary for attaining high energy levels.

QCA and NCA employ distinct criteria for identifying necessary conditions. QCA relies on the diagonal line of a scatterplot as its reference, whereas NCA modifies this approach by translating or rotating the ceiling line to create a reference line with an intercept. This adaptation allows NCA to analyze the necessary conditions of the outcome variable at various specified levels. Consequently, the necessary conditions identified through QCA typically represent a subset of those found via NCA, with NCA often recognizing a broader range of necessary conditions than QCA [55]. This relationship was reaffirmed in findings where the results from fsQCA's necessary conditions analysis appeared as a subset within the broader scope of NCA.

It is important to recognize that in NCA, the term "necessary condition" refers to lower-level antecedent conditions vital for the energy level of the regional digital innovation ecosystem. These include fundamental aspects such as core innovation actors, regulating innovation actors, digital innovation infrastructures, and digital innovation environment. Attention to these aspects is crucial in enhancing the energy level of the ecosystem. On the other hand, fsQCA defines necessary conditions in terms of achieving a certain degree of affiliation, with this study focusing specifically on the conditions required to attain a high energy level. In conclusion, this paper asserts that there is no individual necessary condition that singularly contributes to a high energy level in the regional digital innovation ecosystem. This finding underscores the presence of a complex configuration effect among the various architects involved.

## 4.2 Sufficiency analysis

The sufficiency analysis of configuration involves examining if a set of configurations, comprised of multiple antecedent conditions, forms a subset of the outcome set. This process begins with setting parameters to ensure the explanatory strength of the configurations and to minimize contradictory configurations. The consistency threshold is established at 0.8, the PRI threshold at 0.75, and the case-frequency threshold at 1 [22]. In the subsequent counterfactual analysis stage, due to the absence of definitive evidence and theory regarding the impact

of antecedent conditions on achieving high energy levels, this study hypothesizes that the "presence or absence" of each of the six antecedent conditions could influence both high and non-high energy level [69, 70]. The final stage involves a standardized analysis using fsQCA. This process yields three types of solutions: complex, parsimonious, and intermediate. Core and peripheral conditions within the configuration paths are identified based on the parsimonious and intermediate solutions. The resulting configuration path that influences the energy level of the regional digital innovation ecosystem is then determined, with the analysis results displayed in Table 6.

In our study, we identified three configuration paths leading to high energy levels in regional digital innovation ecosystems, labeled H1, H2, and H3. Conversely, four paths associated with non-high energy levels are labeled N1, N2, N3, and N4. There is no direct correspondence between paths leading to high and non-high energy levels.

For high energy levels, Solution Consistency is 0.974, indicating high reliability. Solution Coverage is 0.741, explaining approximately 74.1% of cases, suggesting substantial explanatory power. For non-high energy levels, Solution Consistency is 0.945, and Solution Coverage is 0.769, indicating strong explanatory capacity.

Table 5 shows that core conditions for all three configuration paths leading to high energy levels are consistent, characterized as core actor-infrastructure type paths. Specifically, Configuration H1 demonstrates that in the absence of regulating innovation actors, a robust core innovation actor combined with comprehensive digital innovation infrastructure can boost the energy level of the regional digital innovation ecosystem. It explains 10.1% of cases and is observed in regions like Tianjin and Chongqing. In recent years, Chongqing has prioritized sectors of the digital economy like integrated circuits, new displays, intelligent terminals, core devices, intelligent networked vehicles, software, and cybersecurity. It aims to build a globally competitive digital industry cluster. By the end of 2022, Chongqing had nurtured 1,900 regulated core enterprises in the digital economy, generating a total added value of 220 billion yuan. As one of the pioneering national experimental zones for digital economy innovation, Chongqing established a national data and computation center, significantly advancing the development of the industrial Internet and the digitization of various industries.

Configurations H2 and H3, while sharing the same core conditions as H1, differ by including an additional antecedent condition, regulating innovation actors. They have Unique

**Table 6. Configuration paths for the energy level of a regional digital innovation ecosystem.**

| Antecedent Conditions | High Energy level | | | Non-High Energy level | | | |
|---|---|---|---|---|---|---|---|
| | H1 | H2 | H3 | N1 | N2 | N3 | N4 |
| Core Innovation Actors | • | • | • | ⊗ | • | ⊗ | ⊗ |
| Collaborative Innovation Actors | | • | • | | ⊗ | ⊗ | ⊗ |
| Digital Governance Actors | • | • | | ⊗ | ⊗ | ⊗ | • |
| Regulating Innovation Actors | ⊗ | • | • | ⊗ | • | • | • |
| Digital Innovation Infrastructures | • | • | • | ⊗ | ⊗ | ⊗ | • |
| Digital Innovation Environment | • | | • | ⊗ | ⊗ | • | ⊗ |
| Raw Coverage | 0.174731 | 0.571909 | 0.606183 | 0.635582 | 0.119048 | 0.132275 | 0.149471 |
| Unique Coverage | 0.100807 | 0.034274 | 0.068549 | 0.541667 | 0.020503 | 0.031746 | 0.055556 |
| Consistency | 0.992366 | 0.968146 | 0.969893 | 0.947732 | 0.923077 | 1 | 1 |
| Solution Coverage | 0.741264 | | | 0.768519 | | | |
| Solution Consistency | 0.973522 | | | 0.944715 | | | |

Note:•or•signifies the presence of the antecedent condition,⊗or⊗ignifies the absence of the antecedent condition, "blank" signifies the presence or absence of this condition is not critical to the outcome;•or⊗signifies a core condition;•or⊗signifies a peripheral condition.

Coverages of 0.034 and 0.069, respectively, and consistencies of 0.968 and 0.970, indicating strong relevance to high energy levels. Representative cases include Zhejiang, Beijing, Jiangsu, Guangdong, and Shanghai.

These regions lead China's digital economy with the highest levels of digitization and advanced marketization of data. Taking Zhejiang as an example, it has fostered numerous digital economy enterprises. Zhejiang boasts a 100% 5G coverage rate in administrative villages and the highest density of 5G base stations nationwide. It hosts national AI Open Innovation Platforms like Alibaba Cloud's "City Brain" and Hikvision's "Video Perception." With 202 data centers, Zhejiang sets a national benchmark in digital infrastructure. Zhejiang University, in collaboration with Alibaba Cloud, operates the "Wisdom Cloud Lab," focusing on cloud computing, AI, 5G, and the Internet of Things. In 2022, Zhejiang's core digital industries contributed 897.7 billion yuan to its economy, leading the national industrial digitization index for three consecutive years.

The analysis of these paths demonstrates the mutual promotion of core innovation actors and infrastructure in varying contexts. Regions can enhance their digital innovation ecosystem capacity by developing high-quality core innovation actors and comprehensive digital infrastructure, supported by collaborative, and regulating innovation actors, conducive digital innovation environments, and competent digital governance. Such diverse combinations of conditions elevate the overall level of the digital innovation ecosystem, forging varied development pathways. Such a combination of diverse edge conditions aligns with the concept of "different paths going in the same direction," thereby elevating the overall level of the digital innovation ecosystem.

The configuration paths for achieving a high energy level in regional digital innovation ecosystems underscore the critical roles of both core innovation actors and digital innovation infrastructures.

According to endogenous growth theory, core innovation actors within a regional digital innovation ecosystem spontaneously absorb and learn from external resources, transforming them into their own innovation momentum. By leveraging their digital innovation capabilities, these actors create a spillover effect that enhances the ecosystem's ability to harness endogenous power and effectively match external R&D resources. Meanwhile, knowledge spillover theory suggests that the enhancement of regional innovation capacity relies not only on substantial investment in local R&D elements but also on the spatial flow of these elements, which brings in external innovation resources. However, in the absence of financial support and knowledge element mobility, most innovation actors within the ecosystem struggle to achieve breakthrough technological innovations. Digital infrastructure, exemplified by cloud platforms, maintains open data circulation channels through advantages such as cross-temporal and spatial information dissemination, data sharing, and low-cost information access. This creates an environment conducive to knowledge spillover, enhances the ease of knowledge acquisition in the region, and promotes the diffusion efficiency of knowledge spillover, thereby driving digital technology innovation. Ultimately, this leads to a high energy level of in the regional digital innovation ecosystem. Resource-based theory posits that a company's sustainable competitive advantage is built on a collection of valuable, rare, and inimitable resources. With robust regional digital infrastructure, the leverage effect of digital technology encourages innovative entities to increase long-term investments in upgrading digital software and hardware. This reduces the uncertainty risks caused by information asymmetry, lowers the marginal cost of knowledge dissemination, and mitigates the spatial and local limitations of knowledge spillover. Ultimately, this enhances the energy level of the regional digital innovation ecosystem.

This aligns with the research findings of Lin and Lu [38], Li [41], and Li and Rao [10], which collectively indicate that within a fully developed digital innovation infrastructure, the willingness of core innovation actors to innovate is a more significant factor in enhancing the energy level of a system than the contributions of other architects within the ecosystem.

For pathways leading to non-high energy levels in regional digital innovation ecosystems, the key findings are as follows: Configuration N1: Ecosystems are unable to achieve high energy levels due to the absence of advanced core innovation actors, digital governance actors, and digital innovation environments, exacerbated by minor deficiencies in regulating innovation actors and digital infrastructures. Configuration N2: A shortage of sophisticated collaborative innovation actors, digital governance actors, and digital innovation environment, combined with slightly insufficient digital infrastructure, prevents the attainment of high energy levels, despite the presence of notable regulating innovation actors and several core innovation actors. Configuration N3: The lack of advanced core innovation actors, collaborative innovation actors, and digital governance actors, along with somewhat inadequate digital infrastructure, hinders the achievement of high energy levels, even with significant regulating innovation actors and a moderate digital innovation environment. Configuration N4: The absence of high-level core innovation actors, collaborative innovation actors, and digital innovation environments ensures that ecosystems cannot reach high energy levels, irrespective of robust regulating innovation actors, certain digital governance actors, and adequate digital innovation environments.

### 4.3 Robustness analysis

In our study, we assessed the robustness of our findings by modifying the consistency criterion. Specifically, the consistency threshold was raised from 0.80 to 0.855, and the PRI value was increased from 0.75 to 0.8. The fsQCA analysis reveals that with both single and overall consistency of antecedent conditions exceeding 0.9, the identified high energy level configuration paths remain consistent. Furthermore, the non-high energy level configuration paths emerge as subsets of the original configurations, as detailed in Table 7. These findings affirm the robustness of the study's results.

**Table 7. Robustness test.**

| Antecedent Conditions | High Energy level | | | Non-High Energy level | | |
|---|---|---|---|---|---|---|
| | H1 | H2 | H3 | N1 | N2 | N3 |
| Core Innovation Actors | • | • | • | ⊗ | ⊗ | ⊗ |
| Collaborative Innovation Actors | | • | • | | ⊗ | ⊗ |
| Digital Governance Actors | • | • | | ⊗ | ⊗ | • |
| Regulating Innovation Actors | ⊗ | • | • | ⊗ | • | • |
| Digital Innovation Infrastructures | • | • | • | ⊗ | ⊗ | • |
| Digital Innovation Environment | • | | • | ⊗ | • | ⊗ |
| Raw Coverage | 0.174731 | 0.571909 | 0.606183 | 0.635582 | 0.132275 | 0.149471 |
| Unique Coverage | 0.100807 | 0.034274 | 0.068549 | 0.556878 | 0.031746 | 0.055555 |
| Consistency | 0.992366 | 0.968146 | 0.969893 | 0.947732 | 1 | 1 |
| Solution Coverage | 0.741264 | | | 0.748016 | | |
| Solution Consistency | 0.973522 | | | 0.955236 | | |

Note:•or•signifies the presence of the antecedent condition,⊗or⊗signifies the absence of the antecedent condition, "blank" signifies the presence or absence of this condition is not critical to the outcome;•or⊗signifies a core condition;•or⊗signifies a peripheral condition.

## 5. Conclusions, implications, and prospects

### 5.1 Conclusions

First, the enhancement of the energy level of regional digital innovation ecosystems critically depends on the architects. In addition to collaborative innovation actors, core innovation actors, digital governance actors, regulating innovation actors, digital innovation infrastructures, and the digital innovation environment are necessary conditions for elevating the regional digital innovation ecosystem to a high energy level (above 60%).

Second, the contribution of a single architect to achieving this high energy level is limited. Core innovation actors and digital innovation infrastructures serve as fundamental conditions with relatively universal effects, while digital governance actors are more foundational in promoting the energy level of regional digital innovation ecosystems.

Additionally, the six antecedent conditions proposed in this study exhibit multiple concurrent causal relationships that require synergistic effects through different combinations of factors. The high-energy and non-high-energy configuration paths of the regional digital innovation ecosystem show a causally asymmetric relationship, indicating that "all roads lead to Rome".

Lastly, the "core actor-infrastructure" pathway can effectively promote the high energy level operation of the regional digital innovation ecosystem. The combination of core innovation actors with high innovation levels and well-developed digital innovation infrastructure not only achieves a high energy level but also compensates for the shortcomings of other antecedent conditions. However, most non-high energy level configuration paths reflect the critical importance of core innovation actors, collaborative innovation actors, digital governance actors, and the digital innovation environment. In the absence of these elements, even with other conditions, the ecosystem will not achieve a high energy level.

### 5.2 Theoretical and practical implications

Firstly, in contrast to previous fragmented studies that focused on individual factors, this study analyzes the interaction between actors in the regional digital innovation ecosystem through the lens of the 'architect' theory. This approach offers theoretical references for elucidating the enhancement of the energy level within the regional digital innovation ecosystem. While most existing research seeks 'architects' within the ecosystem itself, this paper introduces a comprehensive analytical framework encompassing internal architects, external architects, and digital innovation habitats. This framework significantly contributes to the body of knowledge on 'architect' theory within the field of regional innovation ecosystems.

Secondly, while many previous studies have investigated the net effect of key factors on regional innovation ecosystems, the methodologies often employed do not fully acknowledge the fact that "innovation is a complex process requiring the interaction of multiple factors" [71]. Consequently, this study utilizes fsQCA to illustrate how a regional digital innovation ecosystem integrates its internal architects, external architects, and innovation habitats effectively to achieve high energy levels. The findings corroborate the existing notion that high energy levels necessitate a combination of factors within the ecosystem and contribute to demystifying the 'black box' that influences the development of regional digital innovation ecosystems.

Additionally, this study further delineates the causality between sufficiency and necessity. Regarding sufficient conditions, the optimal strategy is aiming for high levels because such conditions ensure the desired outcome (i.e., "if $X$, then $Y$"). Conversely, for necessary conditions, the most effective strategy is to prevent low levels, as the absence of a certain level of

necessary conditions precludes the desired outcome (i.e., "without $X$, there is no $Y$"). For instance, Configuration H3 demonstrates that achieving a high energy level is not crucial for the governance effectiveness of digital governance actors. However, the results of the NCA bottleneck analysis reveal that to attain an 80% energy level, the digital governance efforts of the regional government must reach at least 11.7%; failing to do so means this configuration path will not yield the desired outcome. This finding aligns with the argument by Dul, Hauff and Bouncken [55] that integrating fsQCA with NCA and interpreting it in terms of its degree enhances the precision of results".

Finally, this study offers practical insights into the advancement of regional digital innovation ecosystems at various stages of development. Firstly, all regions must attain the required level of each antecedent condition; failing to do so could result in these conditions becoming bottlenecks, thereby impeding the achievement of high energy levels. Secondly, each region should strategically allocate its resources to the configuration that most appropriately aligns with its unique circumstances. In regions with high energy levels, it is advisable to actively foster core innovation actors and digital innovation infrastructure. Conversely, regions with non-high energy levels may face more constraints in their strategic options. This finding reinforces the notion that "innovation factors vary across regions at different developmental stages" [72].

## 5.3 Managerial insights

Firstly, the high energy level of regional digital innovation ecosystems has been achieved in China's Beijing, Shanghai, Jiangsu, Zhejiang, and Guangdong. However, a significant disparity exists in many central, western, and northeastern provinces. Regions not yet functioning at a high energy level should, on one hand, begin with a "holistic view" approach and utilize the architectural templates of regions operating at a high energy level to devise development policies suited to local conditions. On the other hand, it is vital to boost capital investment in digital R&D, foster the innovative potential of talent, follow the guidance of industrial demand, explore new R&D models that combine independent, collaborative, and integrated innovation in the context of digitalization, overcome the "lock-in effect" of innovation ecosystem development, and establish a differentiated digital innovation ecosystem with enhanced energy levels.

Secondly, the critical role of digital governance actors in the progression of the regional digital innovation ecosystem needs focused attention. The government should provide appropriate policy support tailored to the different stages of digital innovation ecosystem development in the region. At the initial stage of energy level enhancement, emphasis should be placed on developing digital innovation infrastructure, upgrading information infrastructure such as 5G and the Internet of Things supported by advanced information technology, and speeding up the construction of a sharing platform for scientific research and technological development. This will facilitate the evolution of digital innovation infrastructure. At the high-energy level stage, enhanced encouragement and support should be directed towards core innovation actors, urging them to take on leading and coordinating roles in the digital innovation ecosystem, refining the reward distribution mechanism for the transformation of scientific and technological outputs among innovation actors, intensifying interaction and synergy among the architects, and promoting the concentration of digital innovation actors, thereby contributing to the positive development of the digital innovation ecosystem.

## 5.4 Limitations and prospects

On one hand, this study does not address the changes in architects. Owing to the dynamics of the innovation ecosystem, the roles, and behaviors of architects' change, leading to modifications in the role mechanism of the regional digital innovation ecosystem. As a result, future

research would be enriched by adopting a dynamic evolution perspective, utilizing time series analysis to investigate how variations in architects' roles and behaviors influence the evolution of digital innovation ecosystems. On the other hand, this study is limited to data from 30 provinces in mainland China, which may limit the findings' universal applicability. For future research, it would be beneficial to focus the research subject on specific locales such as cities or high-tech zones. Such a strategy would permit a larger sample size and enable a more intricate examination of the complex mechanisms that drive the capacity enhancement of regional digital innovation ecosystems.

## Author Contributions

**Conceptualization:** Rong Huang, Shuai Mao.

**Data curation:** Rong Huang, Shuai Mao.

**Formal analysis:** Shuai Mao.

**Funding acquisition:** Rong Huang, Shuai Mao.

**Methodology:** Rong Huang, Shuai Mao.

**Project administration:** Rong Huang.

**Resources:** Rong Huang.

**Writing – original draft:** Shuai Mao.

**Writing – review & editing:** Rong Huang, Shuai Mao.

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
