## [Decision Letter · Decision Letter 0]

25 Apr 2024

PONE-D-24-05031Enhancing the Energy Level of Regional Digital Innovation Ecosystems: A Configuration PerspectivePLOS ONE

Dear Dr. Mao,

Thank you for submitting your manuscript to PLOS ONE. After careful consideration, we feel that it has merit but does not fully meet PLOS ONE’s publication criteria as it currently stands. Therefore, we invite you to submit a revised version of the manuscript that addresses the points raised during the review process.

We look forward to receiving your revised manuscript.

Kind regards,

Ashfaque Ahmed Chowdhury, Ph.D., FHEA, FIEB

Academic Editor

PLOS ONE

Journal Requirements:

"RH is supported by The Society Community Building Program of Hubei Association of Higher Education(https://www.hahe.org.cn/). Research on the Collaborative Evolution Path of Innovation and Entrepreneurship Education Ecosystem in Local Universities (Grant No.: 2023XD104)"

"This work was supported by the Society Community Building Program of the Hubei Association of Higher Education. Research on the Collaborative Evolution Path of Innovation and Entrepreneurship Education Ecosystem in Local Universities (Grant No.: 2023XD104)"

"RH is supported by The Society Community Building Program of Hubei Association of Higher Education(https://www.hahe.org.cn/). Research on the Collaborative Evolution Path of Innovation and Entrepreneurship Education Ecosystem in Local Universities (Grant No.: 2023XD104)"

Reviewers' comments:

Reviewer's Responses to Questions

**Comments to the Author**

1. Is the manuscript technically sound, and do the data support the conclusions?

Reviewer #1: Partly

Reviewer #2: Yes

2. Has the statistical analysis been performed appropriately and rigorously? 

Reviewer #1: Yes

Reviewer #2: Yes

3. Have the authors made all data underlying the findings in their manuscript fully available?

Reviewer #1: Yes

Reviewer #2: No

4. Is the manuscript presented in an intelligible fashion and written in standard English?

Reviewer #1: Yes

Reviewer #2: Yes

5. Review Comments to the Author

Reviewer #1: I think the author's literature is very interesting, but improving it can further enhance its quality. The specific modification suggestions are as follows:

1. Highlight the research focus and findings in the introduction section

2. Based on existing literature, attempt to propose research hypotheses

3. Further detailed description of the fsQCA method. Recommend this article to the author for learning https://doi.org/10.3390/systems12020053

4. Attempting to explain the hidden meanings of different configuration results using theory

Reviewer #2: Reviewer's Report

The paper titled "Enhancing the Energy Level of Regional Digital Innovation Ecosystems: A Configuration Perspective" explores the mechanisms of "energy level" in regional digital innovation ecosystems. This study, by integrating the "architects" theory, Necessary Condition Analysis (NCA), and Fuzzy-set Qualitative Comparative Analysis (fsQCA), examines the impact of different types of architects and digital innovation habitats on the energy levels of these ecosystems across 30 provinces in China. The main findings indicate that while no single antecedent condition can significantly affect the energy level alone, core innovation actors and digital infrastructures emerge as key factors. The paper identifies multiple concurrent causal relationships among architects and outlines three core configuration paths that drive high energy levels, classifying them as "core actor-infrastructure" driven. The article has its innovative aspects, but it still has shortcomings:

First, the article provides a review and introduction to "regional innovation ecosystems." However, the term changes from "regional innovation ecosystems" to "regional digital innovation ecosystems" lacks clarity. The authors also do not explain what "regional digital innovation ecosystems" are or how they differ from "regional innovation ecosystems." This part should be clearly introduced before Figure 1.

Second, rather than calling Figure 2 a theoretical model, it is more accurately described as the analytical framework of this article, based on the theoretical foundations in Section 2. I believe it should be placed in Section 3 instead of Section 2.

Third, the section on data collection and indicator selection needs to be more detailed and specific. For example, the statement "data from 2018-2020 is chosen for the antecedent conditions" is vague. Specifically, which data from which year was used?

Fourth, the author raised many questions in line 82 that need to be addressed in the conclusion section.

Finally, although the author spends much space introducing and explaining the "Architects" Theory, I do not understand what distinguishes it from the general actors in an innovation ecosystem, or what additional value using the "Architects" Theory provides.

Additional Points:

On page four, the author's mention that "Concurrently, the advent of digital technology has revolutionized traditional economic models, giving rise to innovation actors" might confuse the reader. It is unclear whether the term "giving rise" refers to an increase in the number, type, or capabilities of innovation actors, and why.

In line 89, the statement "Review of the literature reveals that research on the development paths of regional innovation ecosystems generally progresses through three stages" could be ambiguous. It might lead some readers to think it refers to the development of regional innovation ecosystems themselves rather than the evolution of research on them, which is often divided into three stages.

At the end of line 106, it feels necessary to clarify which approach the paper uses.

Any references to literature that are in Chinese should have "(in Chinese)" added after them to indicate the language of the sources.

6. PLOS authors have the option to publish the peer review history of their article (what does this mean?). If published, this will include your full peer review and any attached files.

Reviewer #1: No

Reviewer #2: No

---

## [Author Response · Author response to Decision Letter 0]

3 Jun 2024

Respond to Reviewer #1:

1. Highlight the research focus and findings in the introduction section.

Response: We sincerely thank you for your thorough review and insightful suggestions regarding our paper. After receiving your feedback, we carefully reviewed the introduction section of our manuscript. We recognized that our initial presentation of the research focus was overly broad and that we had omitted a summary of the research findings. Following your recommendations, we have now included detailed descriptions of the research focus and main findings in the introduction section. These revisions aim to clarify the core content of our research. The specific changes are as follows:

The research focus(L134): To address the research gaps mentioned above, this study will establish a framework for analyzing the architects of regional digital innovation ecosystems based on the "architects" theory. It will adopt a configurational perspective and utilize NCA along with fsQCA, which integrates quantitative and qualitative approaches, to analyze 30 provinces (cities, districts) in China. The aim is to explore the multiple simultaneous causal relationships and diverse configurational paths through which different combinations of influencing factors contribute to the elevation of the energy level of regional digital innovation ecosystems.

The research findings (Line142): First, individual architects are not prerequisites for achieving a high energy level in the regional digital innovation ecosystem. However, core innovation actors and digital infrastructures emerge as pivotal conditions. Numerous concurrent causal relationships interconnect core innovation actors, collaborative innovation actors, digital governance actors, regulatory innovation actors, digital infrastructure, and digital innovation environments, forming a complex web of interrelated components.

Second, the study reveals that the "core actor-infrastructure" pathway elevates the energy level of regional digital innovation ecosystems. A synergy between core innovation actors with high innovation levels and well-developed digital innovation infrastructure not only achieves a high energy level but also compensates for deficiencies in other antecedent conditions.

Third, digital governance actors are identified as pivotal bottlenecks influencing the energy level. The results of the NCA demonstrate that the presence of digital governance actors is the sole necessary condition to attain a 10% energy leve. This emphasizes the fundamental role of digital governance actors in boosting the energy level.

Finally, configuration pathways leading to a non-high energy level emphasize the importance of four key elements: core innovation actors, collaborative innovation actors, digital governance actors, and digital innovation environment. Without these, achieving a high energy level is unattainable, regardless of other conditions.

Additionally, to provide readers with a better understanding of the paper's framework, we have included an overview of the article's structure at the end of the introduction (Line 162).

The structure of this article is as follows: Section 2 provides the theoretical foundation and research hypotheses, Section 3 outlines the research methodology, variable selection, and data sources, Section 4 presents and discusses the research findings, Section 5 summarizes the research conclusions and contributions, and provides further research directions.

We hope these revisions adequately reflect the focus and findings of our research. Thank you again for your valuable suggestions, which have helped clarify the direction of our study.

2. Based on existing literature, attempt to propose research hypotheses.

Response: We sincerely appreciate your constructive feedback. In response to your review comments, we have re-examined the impact of architects on the enhancement of energy level of regional digital innovation ecosystem. We reviewed relevant literature and referenced the works of B. Liu et al. (2023), Owen, Vedanthachari, & Hussain (2023), Valavanidis (2020), Castañón & Bustamante (2021), Bejjani, Göcke, and Menter (2023), Makori (2023), Alqasa and Talat (2023), and Vaidian et al. (2022). Based on existing literature, we have proposed several potential research hypotheses to enhance the academic value and practicality of our paper. The specific hypotheses are as follows (Line 295):

Hypothesis 1: Core innovation actors can effectively enhance the energy level of regional digital innovation ecosystems.

Hypothesis 2: Collaborative innovation actors play a significant role in enhancing the energy level of regional digital innovation ecosystems.

Hypothesis 3: Digital governance actors can effectively enhance the governance level of regional digital innovation ecosystems.

Hypothesis 4: Regulating innovation actors play an important moderating and supporting role in the enhancement of the energy level of regional digital innovation ecosystems.

Hypothesis 5: Digital infrastructures provide platform support for the elevation of the energy level of regional digital innovation ecosystems.

Hypothesis 6: The digital innovation environment provides environmental security for the enhancement of the energy level of regional digital innovation ecosystems.

These efforts have allowed us to gain a deeper understanding of the current academic progress on related issues. By proposing research hypotheses, we have not only provided direction for our study but also established a clear framework for data analysis and interpretation of results, thereby enhancing the structure of the paper. Thank you again for your patience and assistance.

3. Further detailed description of the fsQCA method. Recommend this article to the author for learning https://doi.org/10.3390/systems12020053.

Response: We sincerely appreciate your valuable suggestions. Following your advice, we reviewed relevant literature and in the "3.1 NCA and fsQCA" section (Line 369), we referenced the research findings of Dou Z & Sun Y. (2024) to elaborate on the fundamental principles (Line 376). We highlighted that "The fsQCA utilizes a fuzzy membership calibration method to transform data into any numerical representation within the [0-1] interval, effectively avoiding the loss of data information during processing. It can be applied not only to categorical sample data but also to continuous data", and "The fsQCA method can detect multiple concurrent causal relationships and reveal the complex relationships among various actors' influences on the elevation of the energy level of regional digital innovation ecosystems from a holistic perspective", It identifies "which configurations of conditions lead to the occurrence of the outcome variable, and which configurations of conditions lead to the non-occurrence of the outcome variable." Which perfectly aligns with the multi-path research process of elevating the energy level of regional digital innovation ecosystems.

For NCA (Line391), we referenced the findings of Dul, et al. (2023) and added, "NCA tests the necessity of conditional variables by employing two techniques: the Ceiling Envelope with Free Disposal Hull (CE-FDH) and Ceiling Regression with Free Disposal Hull (CR-FDH). This approach not only identifies the necessary conditions leading to the outcome variable from a quantitative perspective but also quantifies the magnitude and bottleneck degree of these necessary conditions."

Thank you once again for your suggestions.

4. Attempting to explain the hidden meanings of different configuration results using theory.

Response: We sincerely appreciate your valuable suggestions. Upon receiving your feedback, we carefully reviewed the manuscript and found that the explanation of the configuration results lacked theoretical support, appearing somewhat vague and hollow. Therefore, following your advice, we made some modifications to the manuscript, attempting to explain the configuration results of the elevation of the energy level of regional digital innovation ecosystems using endogenous growth theory, knowledge spillover theory, and resource-based theory, to enhance the interpretative power of the research results. The modified content is as follows (Line 704):

According to endogenous growth theory, core innovation actors within a regional digital innovation ecosystem spontaneously absorb and learn from external resources, transforming them into their own innovation momentum. By leveraging their digital innovation capabilities, these actors create a spillover effect that enhances the ecosystem's ability to harness endogenous power and effectively match external R&D resources. Meanwhile, knowledge spillover theory suggests that the enhancement of regional innovation capacity relies not only on substantial investment in local R&D elements but also on the spatial flow of these elements, which brings in external innovation resources. However, in the absence of financial support and knowledge element mobility, most innovation actors within the ecosystem struggle to achieve breakthrough technological innovations. Digital infrastructure, exemplified by cloud platforms, maintains open data circulation channels through advantages such as cross-temporal and spatial information dissemination, data sharing, and low-cost information access. This creates an environment conducive to knowledge spillover, enhances the ease of knowledge acquisition in the region, and promotes the diffusion efficiency of knowledge spillover, thereby driving digital technology innovation. Ultimately, this leads to a high energy level of in the regional digital innovation ecosystem. Resource-based theory posits that a company's sustainable competitive advantage is built on a collection of valuable, rare, and inimitable resources. With robust regional digital infrastructure, the leverage effect of digital technology encourages innovative entities to increase long-term investments in upgrading digital software and hardware. This reduces the uncertainty risks caused by information asymmetry, lowers the marginal cost of knowledge dissemination, and mitigates the spatial and local limitations of knowledge spillover. Ultimately, this enhances the energy level of the regional digital innovation ecosystem.

We hope that through these modifications, we can better explain the potential implications of different configuration results and enhance the depth of the research as well as the persuasiveness of the conclusions. Once again, we thank you for your guidance and support. We look forward to your further feedback. 

Respond to Reviewer #2:

First, the article provides a review and introduction to "regional innovation ecosystems." However, the term changes from "regional innovation ecosystems" to "regional digital innovation ecosystems" lacks clarity. The authors also do not explain what "regional digital innovation ecosystems" are or how they differ from "regional innovation ecosystems." This part should be clearly introduced before Figure 1.

Response: Thank you for your valuable feedback. Based on your suggestions, we have made corresponding modifications and provided additional explanations in the manuscript. Now, let me provide detailed explanations of the specific changes:

First, based on the studies of scholars such as Chen & Cai (2023), Li & Rao (2023), and Yi et al. (2023), we have defined the "regional digital innovation ecosystem" (Line 60) as follows: a regional digital innovation ecosystem refers to a complex, dynamic system characterized by the generation, application, and diffusion of digital innovation within a specific spatial and temporal context. This system operates within a digital ecological framework and functions through the competition and cooperation of digital innovation-related actors engaged in value co-creation.

In this definition, constraining the regional digital innovation ecosystem within specific geographical and temporal boundaries implies that its activities and impacts can be identified and analyzed within a defined framework. The term "Digital ecological framework" denotes that the ecosystem operates within a digitalized environment, relying on digital technologies, platforms, and infrastructure such as the internet, cloud computing, and big data. "Competition and cooperation" suggest that actors within the ecosystem engage in both competitive and cooperative relationships, fostering efficiency through competition while facilitating resource sharing and value co-creation through cooperation. "Complex" underscores that the regional digital innovation ecosystem is a complex system comprising numerous interconnected elements and participants, whose operational mechanisms and outcomes are challenging to fully anticipate and manage. "Dynamic" highlights that the ecosystem exhibits dynamic characteristics, continuously evolving with changes in time and space, reflecting the influence of various factors such as technological advancements, shifts in market demand, and policy adjustments. Collectively, these characteristics delineate a regional digital innovation ecosystem grounded in digital technologies and platforms, propelled by the competitive and cooperative interactions of diverse actors to foster innovation.

Second, we have delineated the distinctions between "regional digital innovation ecosystems" and "regional innovation ecosystems" (Line 64): Compared with the traditional regional innovation ecosystem, the regional digital innovation ecosystem is supported by digital technology with convergence characteristics, which facilitates a closer symbiotic relationship of mutual synergy among constituent elements such as innovation actors and innovation resources. Furthermore, the driving effect of regional digital innovation ecosystems on regional innovation development has become more pronounced due to the enabling role and platform effect of digital innovation.

According to the above content, the primary distinction between the regional digital innovation ecosystem and the regional innovation ecosystem lies in technological support. The regional innovation ecosystem predominantly relies on physical infrastructure and traditional technologies, whereas the regional digital innovation ecosystem is supported by convergent digital technologies. Influenced by these different technological supports, regarding collaborative relationships, the regional innovation ecosystem primarily interacts through physical space and traditional methods, resulting in relatively loose collaborative relationships among innovation actors. Conversely, the regional digital innovation ecosystem, facilitated by digital technology, fosters more closely interdependent relationships and cultivates tighter symbiotic relationships among its actors. Regarding driving forces, the regional innovation ecosystem mainly depends on the accumulation and input of traditional industries and offline resources, whereas the regional digital innovation ecosystem achieves faster innovation diffusion and application primarily through digital technology and platforms, exerting a more significant driving force on regional innovation development.

With the above supplements, we aim to enhance the coherence of the paper's logic and the clarity of the research objects. 

Once again, thank you for your thorough review and constructive suggestions on our paper, which have made it more comprehensive and rigorous. We eagerly anticipate your further feedback.

Second, rather than calling Figure 2 a theoretical model, it is more accurately described as the analytical framework of this article, based on the theoretical foundations in Section 2. I believe it should be placed in Section 3 instead of Section 2.

Response: Thank you very much for your constructive suggestions. We have found that renaming "Figure 2" to "Analytical Framework" is indeed more accurate. Following your advice, we have made modifications to the relevant content to better reflect the nature and position of Figure 2. The specific changes are as follows:

Regarding the description of Figure 2, we have revised the title of Figure 2 to "Analytical Framework" (Line 516) to more accurately describe its role in the article.

Regarding the positioning of Figure 2, we have moved Figure 2 from section 2.2 to "

---

## [Decision Letter · Decision Letter 1]

10 Jul 2024

PONE-D-24-05031R1Enhancing the Energy Level of Regional Digital Innovation Ecosystems: A Configuration PerspectivePLOS ONE

Dear Dr. Mao,

Thank you for submitting your manuscript to PLOS ONE. After careful consideration, we feel that it has merit but does not fully meet PLOS ONE’s publication criteria as it currently stands. Therefore, we invite you to submit a revised version of the manuscript that addresses the points raised during the review process.

We look forward to receiving your revised manuscript.

Kind regards,

Ashfaque Ahmed Chowdhury, Ph.D., FHEA, FIEB

Academic Editor

PLOS ONE

Journal Requirements:

Reviewers' comments:

Reviewer's Responses to Questions

**Comments to the Author**

1. If the authors have adequately addressed your comments raised in a previous round of review and you feel that this manuscript is now acceptable for publication, you may indicate that here to bypass the “Comments to the Author” section, enter your conflict of interest statement in the “Confidential to Editor” section, and submit your "Accept" recommendation.

Reviewer #1: All comments have been addressed

Reviewer #2: All comments have been addressed

2. Is the manuscript technically sound, and do the data support the conclusions?

Reviewer #1: Yes

Reviewer #2: Yes

3. Has the statistical analysis been performed appropriately and rigorously? 

Reviewer #1: Yes

Reviewer #2: Yes

4. Have the authors made all data underlying the findings in their manuscript fully available?

Reviewer #1: Yes

Reviewer #2: Yes

5. Is the manuscript presented in an intelligible fashion and written in standard English?

Reviewer #1: Yes

Reviewer #2: Yes

6. Review Comments to the Author

Reviewer #1: The author made revisions based on the review comments and returned to the relevant issues. This research work is worthy of recognition

Reviewer #2: Overall, the paper has seen significant improvement and has effectively addressed the issues I raised in the previous version. The revisions have greatly enhanced the clarity and coherence of the manuscript.

However, from the subheadings, it appears that Section 4 is more focused on analysis and results rather than just the analysis process. I suggest the author review and refine the subheadings to accurately reflect the content of each section. This will help in better organizing the manuscript and guiding the reader through the findings.

7. PLOS authors have the option to publish the peer review history of their article (what does this mean?). If published, this will include your full peer review and any attached files.

Reviewer #1: No

Reviewer #2: No

---

## [Author Response · Author response to Decision Letter 1]

20 Jul 2024

Respond to Reviewer #1:

1.The author made revisions based on the review comments and returned to the relevant issues. This research work is worthy of recognition

Response: Thank you for all your constructive suggestions which improved the quality of our paper. Thank you so much.

Respond to Reviewer #2:

1.Overall, the paper has seen significant improvement and has effectively addressed the issues I raised in the previous version. The revisions have greatly enhanced the clarity and coherence of the manuscript.

However, from the subheadings, it appears that Section 4 is more focused on analysis and results rather than just the analysis process. I suggest the author review and refine the subheadings to accurately reflect the content of each section. This will help in better organizing the manuscript and guiding the reader through the findings.

Response: Thank you very much for your valuable suggestions. In the fourth section of our manuscript, the content indeed includes the results of the analysis, so having "Analysis" as the title alone is neither comprehensive nor accurate. We appreciate your suggestion and have changed the title of the fourth section to "Analysis and results."

Following your advice, we have revisited all the subtitles in the manuscript. We reviewed papers on the PLOS ONE website that adopted similar research methodologies., and referred to the paper structures of scholars such as Huang, Y., Li, K., & Li, P. (2023), Gong, Z., Wang, Y., & Li, M. (2024), Wang, Q., Gao, Y., Cao, Q., Li, Z., & Wang, R. (2023), Miao, Z., & Zhao, G. (2023), and Pappas, I. O., Kourouthanassis, P. E., Giannakos, M. N., & Chrissikopoulos, V. (2016). Based on these references, we have re-examined and modified all the subtitles in our manuscript to better reflect the content of each section.

Additionally, we have adjusted the capitalization of all subtitles (including figure and table captions) according to the journal's submission requirements, hoping to better align with the journal's standards.

The specific adjustments are as follows:

The original fourth section "Analysis" has been changed to "Analysis and results".

The fifth section "Conclusions" has been changed to "Conclusions, implications, and prospects". Specifically, "Concluding Remarks" has been changed to "Conclusions". " Contributions and Practical Implications" has been changed to "Theoretical and practical implications". "Limitations and Future Directions" has been changed to "Limitations and prospects".

The adjusted structure of the manuscript is as follows:

"Section 1, Introduction

Section 2, Theoretical foundation and research hypothesis

Section 3, Materials and methods

Section 4, Analysis and results

Section 5, Conclusions, implications, and prospects"

We hope that these changes accurately reflect the content of the study and help readers quickly grasp the research framework of the paper. Once again, we sincerely thank you for your suggestions, which have made our paper's structure clearer and more reasonable.

---

## [Editor Report · Decision Letter 2]

8 Aug 2024

Enhancing the Energy Level of Regional Digital Innovation Ecosystems: A Configuration Perspective

PONE-D-24-05031R2

Dear Dr. Mao,

We’re pleased to inform you that your manuscript has been judged scientifically suitable for publication and will be formally accepted for publication once it meets all outstanding technical requirements.

Kind regards,

Ashfaque Ahmed Chowdhury, Ph.D., FHEA, FIEB

Academic Editor

PLOS ONE

---

## [Editor Report · Acceptance letter]

13 Aug 2024

PONE-D-24-05031R2 

PLOS ONE

Dear Dr. Mao, 

I'm pleased to inform you that your manuscript has been deemed suitable for publication in PLOS ONE. Congratulations! Your manuscript is now being handed over to our production team.

Kind regards, 

on behalf of

Dr. Ashfaque Ahmed Chowdhury 

Academic Editor

PLOS ONE